# Next-Generation DNA Sequencing-Based Gene Panel for Diagnosis and Genetic Risk Stratification in Onco-Hematology

**DOI:** 10.3390/cancers14081986

**Published:** 2022-04-14

**Authors:** Pablo Gargallo, Merche Molero, Cristina Bilbao, Ruth Stuckey, Estrella Carrillo-Cruz, Lourdes Hermosín, Olga Pérez-López, Antonio Jiménez-Velasco, Elena Soria, Marián Lázaro, Paula Carbonell, Yania Yáñez, Iria Gómez, Marta Izquierdo-García, Jennifer Valero-García, Carlos Ruiz, Esperanza Such, Inés Calabria

**Affiliations:** 1Health In Code Group, Oncology Department, 46980 Paterna, Spain; merche.molero@healthincode.com (M.M.); marian.lazaro@healthincode.com (M.L.); paula.carbonell@healthincode.com (P.C.); yania.yanez@healthincode.com (Y.Y.); iria.gomez@healthincode.com (I.G.); marta.izquierdo@healthincode.com (M.I.-G.); jennifer.valero@healthincode.com (J.V.-G.); carlos.ruiz@healthincode.com (C.R.); ines.calabria@healthincode.com (I.C.); 2Servicio de Hematología, Hospital Universitario de Gran Canaria Dr. Negrín, 35010 Las Palmas de Gran Canaria, Spain; bilbaocristina@gmail.com (C.B.); rstuckey@fciisc.es (R.S.); 3Servicio de Hematología, Hospital Universitario Virgen del Rocío, 41013 Sevilla, Spain; estrellam.carrillo.sspa@juntadeandalucia.es (E.C.-C.); elena.soria@juntadeandalucia.es (E.S.); 4Instituto de Biomedicina (IBIS/CSIC/CIBERONC), Universidad de Sevilla, 41013 Sevilla, Spain; 5Hematology Department, Hospital de Jerez, Carr Madrid-Cádiz, 11407 Jerez de la Frontera, Spain; marial.hermosin.sspa@juntadeandalucia.es; 6Hematology Department, Hospital Universitario Virgen Macarena, 41009 Sevilla, Spain; olgaplopez@gmail.com; 7Servicio de Hematología y Hemoterapia, Hospital Regional Universitario de Málaga, IBIMA, 29010 Málaga, Spain; antoniof.jimenez.sspa@juntadeandalucia.es; 8Department of Hematology, Hospital Universitario y Politécnico La Fe, 46026 Valencia, Spain; such_esp@gva.es; 9Hematology Research Group, Department of Medicine, La Fe Health Research Institute, University of Valencia, 46026 Valencia, Spain

**Keywords:** acute myeloid leukemia, myelodysplastic syndrome, myeloproliferative neoplasms, acute lymphoblastic leukemia, myeloid neoplasms with germline predisposition, NGS panel, targeted capture sequencing

## Abstract

**Simple Summary:**

The present work comes up after detecting stakeholders’ need to test for manifold molecular biomarkers in each sample from an individual diagnosed with myeloid neoplasm or acute leukemia. The development of gene panels based on NGS technology is considered a potentially effective testing alternative with less human effort, compared to that required by other conventional techniques (PCR, FISH, conventional karyotype, etc.). The validation of this panel aims to propose a new solution for hospitals to face the challenges posed by the molecular study of this group of onco-hematological diseases.

**Abstract:**

A suitable diagnostic classification of myeloid neoplasms and acute leukemias requires testing for a large number of molecular biomarkers. Next-generation sequencing is a technology able to integrate identification of the vast majority of them in a single test. This manuscript includes the design, analytical validation and clinical feasibility evaluation of a molecular diagnostic kit for onco-hematological diseases. It is based on sequencing of the coding regions of 76 genes (seeking single-nucleotide variants, small insertions or deletions and CNVs), as well as the search for fusions in 27 target genes. The kit has also been designed to detect large CNVs throughout the genome by including specific probes and employing a custom bioinformatics approach. The analytical and clinical feasibility validation of the Haematology OncoKitDx panel has been carried out from the sequencing of 170 patient samples from 6 hospitals (in addition to the use of commercial reference samples). The analytical validation showed sensitivity and specificity close to 100% for all the parameters evaluated, with a detection limit of 2% for SNVs and SVs, and 20% for CNVs. Clinically relevant mutations were detected in 94% of all patients. An analysis of the correlation between the genetic risk classification of AML (according to ELN 2017) established by the hospitals and that obtained by the Haematology OncoKitDx panel showed an almost perfect correlation (K = 0.94). Among the AML samples with a molecular diagnosis, established by the centers according to the WHO, the Haematology OncoKitDx analysis showed the same result in 97% of them. The panel was able to adequately differentiate between MPN subtypes and also detected alterations that modified the diagnosis (*FIP1L1*-*PDGFRA*). Likewise, the cytogenetic risk derived from the CNV plot generated by the NGS panel correlated substantially with the results of the conventional karyotype (K = 0.71) among MDS samples. In addition, the panel detected the main biomarkers of prognostic value among patients with ALL. This validated solution enables a reliable analysis of a large number of molecular biomarkers from a DNA sample in a single assay.

## 1. Introduction

Molecular characterization of hematological malignances currently enables a comprehensive diagnostic and prognostic classification [1] and can even be translated into therapeutic recommendations in some cases [2]. The number of genetic biomarkers with clinical implications is constantly growing, and the latest World Health Organization (WHO) update for myeloid neoplasms and acute leukemia, published in 2016, already includes a substantial number of them [3]. The most frequently disrupted genes in each of these pathologies have been reasonably well defined during the last decade. Regarding *de novo* acute myeloid leukemia (AML), about 25 genes are recurrently mutated. The most commonly altered gene is *FLT3* (28%), followed by *NPM1* (27%) and *DNMT3A* (26%). Among fusions, the *PML*-*RARA* in-frame rearrangement is identified in up to 9% of all samples, being the most prevalent [4]. The prevalence of genetic alterations in AML varies in different series, depending on the disease subtype analyzed and the clinical context of the patients included in the studies [5,6]. Overall, the most mutated genes in MDS are *TET2* (20–25%), *DNMT3A* (12–18%), *ASXL1* (15–25%) and *SF3B1* (20–30%) [7,8]; up to 30 genes are commonly altered [7,8]. Furthermore, myeloproliferative neoplasms are divided molecularly into two large groups, namely, *BCR-ABL1*-positive chronic myeloid leukemia and Philadelphia chromosome-negative neoplasms. In the latter group, the three most clinically relevant genes analyzed in routine clinical care are *JAK2*, *CALR* and *MPL* [9]. The prevalence of mutations in each one depends on the entity under consideration (polycythema vera, essential thrombocythemia or primary myelofibrosis) [9]. Finally, in the context of acute lymphoblastic leukemia (ALL), the mutational landscape is subject to the lineage (B or T) as well as to the age of the patients. The most widespread genetic alteration in adulthood ALL is the disruption of *IKZF1* (25–35% of cases), while it is altered in 12–17% of pediatric patients. In children, the *ETV6-RUNX1* fusion is the most common rearrangement (22%), while in adults it is the *BCR-ABL1* fusion (identified in 25% of cases) [10]. One of the main challenges when studying the molecular profile of these hematological malignancies is the wide range of mandatory routine genetic analysis, including single-nucleotide variants (SNVs), copy number variants (CNVs), chromosomal rearrangements (SVs), as well as other structural changes, such as tandem duplications or variations in cell ploidy [3,7,10,11,12,13,14]. This requires the performance of multiple techniques, such as the conventional karyotype, FISH, as well as different PCR approaches, some of them specific to each disease. Therefore, a number of different molecular techniques must be implemented in hospitals for an accurate characterization of hematological malignances, which is less cost-effective as the number of recommended biomarkers increases [15].

Meanwhile, the rapid development of next-generation sequencing (NGS) technologies has boosted their implementation in daily clinical practice. The feasibility of NGS has been proved in different onco-hematological diseases, especially in the field of acute myeloid leukemia [12]. NGS solutions provide a comprehensive approach to different biomarkers in a single assay. However, the detection of such a large number of genetic biomarkers of different natures is still a technological challenge. One of the main challenges of NGS lies in obtaining a CNV analysis that would correlate with conventional karyotype or SNP array. Numerous efforts are being made in this sense, and the first results are already available [16].

Beyond obtaining reliable results at an affordable price, it is also important to generate an amount of data manageable by the stakeholders. Following this premise, sequencing the whole genome or exome generates a large amount of information, and most of these data may be misleading or useless for patient management in daily clinical practice.

Certainly, implementing NGS-based gene panels in clinical practice raises several challenges as well. Some of them are inherent to NGS technologies and could be solved with experience and knowledge. However, two particular challenges hamper the implementation of NGS solutions in the daily clinical practice of onco-hematology laboratories. First, the current international guidelines are based on previous experience; therefore, NGS, due to a lack of standardization, is not the main technology recommended for analyzing several biomarkers. Second, there are insufficient cost efficiency studies in this clinical field that would demonstrate the advantage of implementing NGS. Despite these challenges, NGS-based gene panels enable the identification of target genetic alterations in a single sample and in a single assay. Therefore, it may be a useful strategy for molecular diagnosis in onco-hematology that could replace or complement the conventional techniques.

Accordingly, we developed a DNA NGS-based gene panel that focuses on identifying the genetic markers recommended by the main clinical guidelines [3,4,5,6,7,8,9]. The main objective of the study was to validate the sensitivity, specificity, repeatability and reproducibility of the Haematology OncoKitDx for the detection of clinically actionable variants, as well as to explore its clinical feasibility and utility.

## 2. Materials and Methods

### 2.1. Panel Design

The panel targets the main biomarkers for diagnosis, prognosis, therapy and resistance among the most frequent onco-hematological diseases, namely AML, myeloproliferative neoplasm (MPN), myelodysplastic syndrome (MDS) and ALL. It was designed according to the current World Health Organization classification [3], the ELN (European Leukemia Network) working group recommendations [11], the NCCN guidelines [7,10,13,14] and the Food & Drug Administration and European Medicines Agency drugs approval. 

The DNA NGS-based panel Haematology OncoKitDx examines the coding regions of 78 genes by complete sequencing, looking for several types of genetic alterations, such as point mutations and small deletions and insertions. The included genes are *ARID5B*, *ASXL1*, *ASXL2*, *ATRX*, *BCOR*, *BCORL1*, *BLNK*, *BRAF*, *CALR*, *CBL*, *CDKN2A*, *CDKN2B*, *CEBPA*, *CHIC2*, *CREBBP*, *CRLF2*, *CSF3R*, *CSNK1A1*, *CUX1*, *DDX3X*, *DDX41*, *DNMT3A*, *EP300*, *ETNK1*, *ETV6*, *EZH2*, *FBXW7*, *FLT3*, *GATA1*, *GATA2* (intron 4 included), *GATA3*, *HAVCR2*, *IDH1*, *IDH2*, *IKZF1*, *IL7R*, *JAK1*, *JAK2*, *JAK3*, *KIT*, *KMT2A*, *KMT2C*, *KRAS*, *MPL*, *NF1*, *NFE2*, *NOTCH1*, *NPM1*, *NR3C1*, *NRAS*, *P2RY8*, *PAX5*, *PHF6*, *PIGA*, *PPM1D*, *PTEN*, *PTK2B*, *PTPN11*, *RAD21*, *RB1*, *RUNX1*, *SETBP1*, *SF3B1*, *SH2B3*, *SMC1A*, *SMC3*, *SRP72*, *SRSF2*, *STAG1*, *STAG2*, *STAT5B*, *TET2*, *TP53*, *TPMT*, *TYK2*, *U2AF1*, *WT1* and *ZRSR2.* A low-density SNP array of 996 single-nucleotide polymorphisms (SNPs) located throughout the entire genome was included in the design of the panel, and a custom bioinformatics pipeline was developed with the purpose of detecting CNVs throughout the entire genome, including gain and loss of whole chromosomes, chromosomal arms, genes or even exons. In addition, the panel was also designed to identify rearrangements by detecting the intronic breakpoints previously described in 27 genes: *ABL1*, *ABL2*, *BCR*, *CBFA2T3*, *CBFB*, *CSF1R*, *EPOR*, *ETV6*, *FGFR1*, *FUS*, *JAK2*, *KMT2A*, *MEF2D*, *MNX1*, *MYH11*, *NPM1*, *NUP214*, *NUP98*, *PDGFRA*, *PDGFRB*, *RARA*, *RBM15*, *RUNX1*, *SET*, *STIL*, *TAL1* and *TCF3.* This strategy makes it possible to detect fusions of these genes with any other partner throughout the genome. The Haematology OncoKitDx also tests for 14 relevant SNPs in pharmacogenetics according to the PharmGKB database (2A evidence or higher) [17].

### 2.2. Sample Selection, Preparation, Sequencing, Bioinformatics Pipeline and Variant Classification

A total of 63 samples were selected for the analytical validation of the assay: 9 reference samples from Coriell Cell Repositories (3), Horizon Discovery (2), Incliva Biobank (2) and Agilent Technologies (2); and a total of 54 bone marrow samples from patients. Sensitivity and specificity for point variations and small indels detection were assessed in 19 samples. Of these, 12 samples were employed for CNV evaluation, 19 samples for fusions and large rearrangements analysis, 3 for pharmacogenetics testing and 11 samples for *FLT3*-ITD detection analysis. Both repeatability and reproducibility evaluation were conducted with samples previously analyzed in this technical validation.

In order to evaluate the assay’s feasibility in the clinical context, 170 bone marrow DNA samples were assessed for SNVs, INDELs, CNVs and fusions. Samples from six Spanish hospitals were selected based on their own scientific or clinical interest. Most samples (153/170) had been previously analyzed with different conventional techniques (RT-qPCR, FISH or karyotype) and/or by means of other commercial NGS panels (Sophia myeloid solution, Oncomine Myeloid Research Assay, Oncomine Childhood Cancer Research Assay or TruSight Myeloid Sequencing Panel). 17 of the 170 samples had not been previously studied by any molecular technique but were also included.

DNA sample concentration was measured by fluorometric quantification using the Qubit dsDNA BR Assay kit (Thermo Fisher Scientific, Waltham, MA, USA) and the Qubit dsDNA HS Assay kit (Invitrogen, Waltham, MA, USA). A dilution with 50–100 ng nuclease-free water in 7 µL was prepared to start the library preparation protocol. Enzymatic fragmentation of 50–100 ng of gDNA from each sample to an average of 175 bases was carried out with the SureSelect XT HS and XT Low input enzymatic fragmentation kit (Agilent). The library preparation was automated by means of the Magnis NGS Prep System and Magnis SureSelect XT HS kit (Agilent). To obtain indexed libraries by molecular barcoding, unique molecular indexes (UMIs) were employed and the Agencourt AMPure XP beads (Beckman Coulter, Brea, CA, USA) were used for purification. Libraries were universally amplified by PCR (9 or 10 cycles, depending on the amount of input DNA) with adapter-specific primers for the union of indexes, universal indexes and sequencing adapters. Indexed DNA fragments were then purified and library quality control was performed on the Agilent 2200 TapeStation. A mix of biotinylated capture probes, specifically designed for the Haematology OncoKitDx, was hybridized to regions of interest and then captured with streptavidin beads. Library enrichment involved post-capture PCR amplification (12 cycles) and purification with beads. Libraries were quantified using the Qubit dsDNA HS Assay kit (Invitrogen). A denaturation protocol was carried out prior to sequencing, using the NextSeq 500/550 Mid Output v2.5 and NextSeq 500/550 High Output v2.5 sequencing kits (Illumina) and Phix control (Illumina). Libraries were diluted to 1.5 pM. Pools were then loaded onto the NextSeq 550 system (Illumina) for massive library sequencing in “Stand-alone” mode with 2 × 75 paired-end reads following the manufacturer’s instructions. The bioinformatics pipeline was set up as previously reported by Martinez-Fernández P et al. [18].

Classification of the identified variants using the DataGenomics software was done according to their functional and clinical evidence, as described by Martinez-Fernández P et al. [18].

The assay’s analytical validity was also evaluated by establishing the percentage of samples carrying variants with diagnostic, prognostic and therapeutic implications according to guidelines (tier I variants) and high-impact literature (tier II variants) (Tier I and II biomarkers stated in Appendix A). This percentage of samples carrying clinically relevant variants was established based on the identification of SNVs, SVs and gains and losses in target genes, but not on the presence of large CNVs. Large CNVs were considered clinically relevant in all cases. The percentage of clinically relevant variants were grouped into each of the four parameters of interest (diagnosis, prognosis, therapy and resistance), for each neoplasm aims to describe and summarize the collected data in a global way. As the panel includes the most frequently mutated genes and genes with the highest level of clinical evidence for each of the pathologies analyzed, the number of variants with clinical evidence at the diagnostic, prognostic, therapeutic and resistance-to-therapy levels is expected to be high and consistent with previous reports. This description is not intended to create the impression that the panel detects something other than what might be expected, based on the knowledge available on the analyzed pathology; it is simply to illustrate how the biomarkers that are expected to be found may be detected by a single test, in this case, by the NGS test.

### 2.3. Clinical Feasibility Evaluation

The genetic risk assigned by the Haematology OncoKitDx, according to the ELN-2017 recommendations for AML patients, was compared with the risk originally assigned by its reference hospital after performing a conventional karyotype and a commercial NGS panel. The patients were classified into three categories (Favorable, Intermediate and Adverse) based on the genetic risk according to ELN [4]. Two extra categories were added (Favorable or Intermediate; Intermediate or Adverse) to classify those cases in which the *FLT3* allelic ratio could not be used to establish the definitive genetic risk group. A Cohen’s Kappa test was performed to establish the correlation between both approaches. This test quantifies the agreement between observers that independently classify the same n units into the same k nominal or ordinal categories. It adjusts the observed proportion of agreement and ranges from −P^c^/(1 − P^c^) to 1, where P^c^ is the expected agreement that results from chance. The strength of agreement in the Kappa statistic is <0 Poor; 0–0.2 Slight; 0.2–0.4 Fair; 0.4–0.6 Moderate; 0.6–0.8 Substantial; 0.8–1 Almost Perfect; 1 Perfect [19]. Regarding myelodysplastic syndromes, the cytogenetic risk, assigned according to the IPSS-R guidelines by means of conventional karyotype, was compared with the cytogenetic risk assigned by the Haematology OncoKitDx panel. The level of agreement was established by Cohen’s Kappa [19]. 

Furthermore, we established a diagnosis for AML samples (according to the 2016 WHO classification) based on the Haematology OncoKitDx results. The AML samples were divided into two groups: (1) samples with information about the diagnosis established at their centers or (2) samples referred as AML for which the precise diagnosis according to WHO 2016 established at their hospitals was not provided. 

In the first group, the correspondence between the diagnosis of the center and that derived from Hematology OncokitDx was evaluated. The possible results were match or not match.

In the second group of samples, a diagnosis was assigned according to the current WHO classification based on molecular information derived from the Hematology OncoKitDx panel. The agreement between the diagnosis established by the different hospitals and the result of Haematology OncokitDx could not be assessed. Additionally, to evaluate the clinical usefulness of the panel in this group of patients, it was ascertained whether the molecular alterations detected were more consistent with the suspected diagnosis (AML) than with MDS, MPN, MDS/MPN or myeloid/lymphoid neoplasms.

To consider a molecular alteration or profile of alterations to be more consistent with AML versus MDS, MPN, MDS/MPN or myeloid/lymphoid neoplasms, at least one identified mutation had to meet one of these criteria:
-One or more research manuscripts from Q1 or Q2 journals (JCR 2021) reporting that the mutation is more common in AML than in other myeloid neoplasms (or it is typical of transformation to AML).

or

-One or more clinical guidelines stating that the mutation is more frequent in AML than in other myeloid neoplasms (or it is typical of transformation to AML).

or

-One or more reviews from Q1 or Q2 journals (JCR 2021) claiming that one mutation is more usual in AML than in other myeloid neoplasms (or it is typical of transformation to AML).

or

-Identifying a mutation that is pathognomonic for AML or exclusively described in AML.

Likewise, for the group of samples referred to as myeloproliferative neoplasms, the ability of the panel to differentiate the neoplasms between those *BCR-ABL* positive vs. *BCR-ABL* negative was evaluated. Additionally, in samples negative for *BCR-ABL*, it was analyzed whether the molecular alterations observed were consistent with MPN or, contrarily, were suggestive of MDS or MPN/MDS syndromes.

Finally, due to the small size of the ALL group cohort, it was decided to report the data with diagnostic and prognostic implications in a descriptive manner. The ability to assign a cytogenetic risk in B-ALL through the Haematology OncoKitDx was assessed as per the NCCN guidelines [10].

Concerning the germline, a potential constitutional origin of variants with a VAF (variant allelic frequency) between 35% and 100% in the *SRP72*, *CEBPA*, *DDX41*, *RUNX1*, *ETV6*, *GATA2*, *SRP72* and *TP53* genes was reported. Likewise, possible CNVs in the chromosomal region 14q32.2 were investigated in pursuit of genomic duplications, including both *ATG2B* and *GSKIP* genes. We also sought for mutations in other genes with germline implications not, however, included in the WHO 2016 classification, such as *NF1* or *PTPN11*.

## 3. Results

### 3.1. Analytical Validation

#### 3.1.1. Analytical Performance Indicators

Regarding next-generation sequencing quality scores, the obtained values were 92.9% for Q30 and 90.0% for the average percentage of clusters passing filter. The mean coverage achieved was 1400×. Further, 98.41% of the bases were covered at >20% of the average coverage, allowing for good uniformity, and 97.7% coverage was achieved with a depth of 200×.

The analytical sensitivity and specificity were calculated. Concerning SNV and indel variants, 51 variants were assessed, including *FLT3*-ITDs. Variants were also analyzed by means of other commercial kits based on other techniques, such as fragment analysis, real-time PCR or NGS. All variants in the target genes included in Haematology OncoKitDx were detected. For the CNV analysis, 16 CNVs were evaluated in 12 samples previously analyzed with a different NGS panel, MLPA or digital PCR (dPCR). Concordance of results determined a sensitivity of 100%. Regarding structural rearrangements, 35 known fusions were screened for in 19 samples previously analyzed by real-time PCR, dPCR or conventional PCR. Concordance of results determined a 100% sensitivity and over 99.9% specificity. In the three commercial peripheral blood samples (HapMap) from the Coriell Institute, 12 out of all the pharmacogenetic positions included in the Haematology OncoKitDx were analyzed. All positions were confirmed with 100% analytical sensitivity and specificity greater than 99.9%. 

To assess repeatability, one diluted sample was sequenced in duplicate. The same 163 SNVs were reported in both samples, giving a concordance of 100%. Three discordant variants were excluded from the analysis for being likely an artifact due to their proximity to a homopolymer. Regarding CNVs, results were concordant in both replicates analyzed. CNVs of size less than 0.5 Kb or Low-Quality Score events were excluded from the analysis. In addition, those identified on sex chromosomes X and Y were also excluded for possible imbalance. For large rearrangements, poor-quality events and those with coverage lower than 10 reads were excluded from the analysis. In the samples, one deletion was detected, whose coverage, mate and split reads were consistent. The pharmacogenetic positions were studied and correctly detected in both replicates. 

To establish reproducibility, five samples were analyzed in two different runs. Reagents from different batches were used and the technical staff involved was also different. Those variants detected by the Haematology OncoKitDx panel with a variant allelic frequency greater than 2% that had passed the PASS and d100 (depth higher than 100 reads) quality filters were analyzed by the DataGenomics software. A total of 21 variants were excluded because they were close to homopolymer regions. The reproducibility obtained for SNVs was 98.5%. CNVs detected with a low or medium quality and with a size smaller than 0.5 Kb have not been included. Both duplicates contained three high-quality events in only one sample. For SVs, all events detected in the five samples were coherent in both duplicates. All the pharmacogenetic positions were consistent in both tests for each of the samples. Reproducibility for CNVs, SVs and pharmacogenetics was greater than 99.9%.

The limit of detection (LOD) was established at 2% for SNVs and small deletions/insertions. A sample with 1594 mutations matching genes included in the panel was diluted in negative controls of similar genomic quality to obtain 35 variants at 2% allelic frequency. All variants were detected at a frequency close to the expected VAF, with an average deviation of 0.77 from the expected frequency. For SVs, three samples with known SVs were selected to validate the LOD. Previously, these SVs were specifically amplified and quantified by dPCR versus an endogenous gene (β-globin) to know how represented each SV in the corresponding sample was. Taking into account these percentages, in silico dilutions of the selected samples in a negative control were prepared to obtain the events at different percentages. All analyzed variants were detected with a 2% VAF with High Quality Score. Four samples with known CNVs were selected to validate the LOD for CNV detection. As for SVs, samples with CNVs were previously amplified and quantified by dPCR versus an endogenous gene (β-globin) to know how represented each CNV in the corresponding sample was. Considering these percentages, in silico dilutions of the selected samples in a negative control were prepared to obtain the events at different percentages. LOD was determined considering the total copies of the event in a given sample and established at 20% for both copy number losses and gains.

Finally, the optimal number of samples to be sequenced per run was calculated to guarantee a coverage of 97.7% and a depth of 200× or 99.3% with a depth of 100×, 8 being the recommended number of samples per run using the NextSeq 500/550 Mid Output v2.5 kit and 24 samples for the NextSeq 500/550 High Output v2.5 kit, allowing a minimum of passing filter (PF) clusters of approximately 17.5 million per sample (Figure 1).

#### 3.1.2. Actionable Genetic Variants (Tier I and Tier II Mutations)

All 170 bone marrow samples met the quality acceptance criteria and were included in one of the four established cohorts, namely acute myeloid leukemia (*n* = 98), myelodysplastic syndrome (*n* = 34), myeloproliferative neoplasm (*n* = 25; including 15 chronic myeloid leukemia) and acute lymphoid leukemia (*n* = 13; 4 samples from adults and 9 pediatric samples). 

It was possible to identify clinically relevant mutations (Tier I or II alterations) in 94% of total patients (160/170), considering exclusively SNVs, SVs and gene gains and losses. Namely, 81.8% of all patients presented at least one variant of prognostic significance (PI or PII) and 78.2% of patients carried variants involved in diagnosis (DI, DII). Additionally, variants with therapeutic relevance were identified in 37.1% of total samples (TI, TII) and 30% of patients presented variants associated with resistance to certain drugs (RI, RII).

Regarding newly diagnosed AML patients (*n* = 98), prognostic variants were identified in 95.9% of total cases, variants with implications for diagnosis in 83.7%, while variants with therapy and resistance association in 40.8% of cases. Concerning MDS patient samples studied at diagnosis (*n* = 34), variants of prognostic value were detected in up to 85.3% of patients. Variants of diagnostic interest were found in 70.6%, therapeutically actionable variants in 11.8% and, finally, resistance to therapy mutations in 20.6% of the samples. Among MPN patients (*n* = 25), the most frequent genetic variants were those of diagnostic significance (76% of the samples). Therapeutically useful mutations were found in 72% of MPN samples, informative prognostic variants in 16% of them and resistance alterations exclusively in 8% of patients. Lastly, among ALL samples (*n* = 13), 84.6% were found to harbor variants related to prognosis. Variants involved in diagnosis were identified in 61.5% of samples and both resistance- and therapy-related variants were detected in 15.4% of ALL patients (Appendix A).

### 3.2. Clinical Feasibility

#### 3.2.1. Genetic Alterations in Acute Myeloid Leukemia

The Haematology OncoKitDx identified *NRAS/KRAS* gene mutations in 44% of AML patients. *FLT3* gene mutations were detected in 29% of cases. The *ASXL1* gene was mutated in 28% and the *DNMT3A* gene in 22%. Other commonly mutated genes among the analyzed AML samples were *TP53* (19%), *TET2* (19%), *NPM1* (17%), *IDH1/2* (14%), *SRSF2* (14%), *RUNX1* (14%), *SF3B1* (9%), *CEBPA* (8%) and *KIT* (3%). Regarding rearrangements included in the WHO 2016 classification and ELN 2017 genetic risk groups, the *CBFB-MYH11* gene fusion was present in 7% of cases and was the most frequently identified rearrangement. Other important gene fusions detected were *RUNX1-RUNX1T1* (2%) and *MLLT3-KMT2A* (1%). Only one *KMT2A* rearrangement other than *MLLT3-KMT2A* was detected (*KMT2A-MLLT10*). No AML cases were identified carrying the *DEK-NUP214*, *BCR-ABL1* or *RBM15-MKL1* gene fusions. 

Integrating the genetic SNVs, SVs and the large CNVs identified, it was possible to suggest a diagnosis according to the WHO 2016 classification and a risk group in agreement with the ELN 2017 guidelines in all 98 AML patient samples (Appendix A).

For 33 of 98 AML cases, the genetic results obtained by means of other commercial NGS panels, as well as from the conventional karyotype, were available and could be provided by the hospitals. The ELN 2017 risk group assigned by the different hospitals was contrasted with the risk group obtained by the Haematology OncoKitDx panel. In 5 of the 33 patients (15%), the karyotype result provided by their hospitals was inconclusive and a risk group could not be assigned. The Haematology OncoKitDx panel was able to assign a risk group in these five cases: N-4, M-7, M-11, M-12, V-16 (Appendix A). In 27 of the remaining 28 patients (96.4%), the risk group assigned by the reference hospital coincided with the one assigned by Haematology OncoKitDx (Appendix A). In one of the cases (N-22; 3.6%), the Haematology OncoKitDx panel assigned an adverse risk according to the ELN 2017 guidelines while the hospital had established an intermediate risk group. This discrepancy was due to the fact that the NGS panel used by the hospital did not detect two mutations associated with adverse risk (in *ASXL1* and *RUNX1*), while they were identified by the Haematology OncoKitDx. Considering these 28 patients, the Cohen’s Kappa Tests obtained a K = 0.94. The strength of agreement between these two approximations was considered almost perfect (Appendix A).

Some comparative examples between the conventional karyotype reported by the hospital and the CNV plot obtained using the Haematology OncoKitDX panel are shown in Figure 2.

In addition, the molecular diagnosis based on the WHO 2016 classification established by the hospital was communicated in 39 AML cases (39/98; Group 1). The same diagnosis was established by means of sequencing Haematology OncoKitDx in 38 of these 39 cases (match 97.4%—not match 2.6%). The only difference was evidenced when analyzing sample F-1, whose diagnosis provided by the center was AML with inv(3)(q21.3q26.2) or t(3;3)(q21.3;q26.2); *GATA2*,*MECOM*(EVI1). However, the first version of the Haematology OncoKitDx panel presented in this work was not able to detect said molecular event, thus, being an expected disagreement (Appendix A).

On the other hand, in 57 of the 98 AML cases the WHO diagnosis established was not transferred from the centers (group 2). A molecular result consistent with AML was obtained in all these samples (100%). Meanwhile, 44 of 57 samples (77.2%) were carriers of alterations more suggestive of AML than of another hematological neoplasm, based on the established criteria reported above (Appendix A).

#### 3.2.2. Genetic Alterations in Myelodysplastic Syndromes

A total of 34 samples corresponding to myelodysplastic syndromes were analyzed. The most frequently mutated gene was *ASXL1*, altered in 31% of patients, followed by *TET2* (28%), *SF3B1* (17%) and *U2AF1* (17%). The prevalence of variants in genes related to MDS detected by Haematology OncoKitDx is shown in Table 1. 

Considering the CNVs, the cytogenetic risk assigned by conventional karyotype according to the IPSS-R classification was compared with that obtained from the CNV analysis by Haematology OncoKitDX. For 17 of 34 MDS samples, the karyotype was available. In 82.4% of these cases, the IPSS-R cytogenetic risk group assigned by conventional karyotype matched the one derived from the CNV analysis. In the remaining 17.6%, a disparity was observed in the risk group assigned (Table 2). The Cohen’s Kappa coefficient was K = 0.71 (Appendix A).

In patient number 3, a balanced translocation detected by conventional karyotype was not observed by NGS, neither as a CNV nor as a rearrangement. In patient number 6, the derived chromosome 6, resulting from the event t (3;6) (q11;p25), was detected as a single event per conventional karyotype, while Haematology OncoKitDX’s CNV analysis counted it as two events on chromosome 6p (6p gain and 6p deletion). This led to a disparity in the assigned cytogenetic risk group, since the number of total cytogenetic events was not the same. Finally, in patient number 17, two CNVs not detected by conventional karyotype were observed by NGS, which also affected the final cytogenetic risk classification.

#### 3.2.3. Genetic Alterations in Myeloproliferative Neoplasm

In the 25 samples submitted as MPN, 15 had been diagnosed with CML and the remaining 10 were submitted as Philadelphia-negative MPN. We identified 15 cases carrying the *BCR-ABL1* fusion and 10 cases that were negative for it, as had been expected and in line with the information provided by the hospitals.

In the 15 samples with a diagnosis of chronic myeloid leukemia provided by the hospitals, we identified fusions involving the *ABL1* gene but, in addition to the *BCR-ABL* rearrangements, we also detected mutations in regulatory genes (*CBL*, *TET2*, *EZH2* and *ASXL1*), RAS signaling genes (*NRAS*, *KRAS*), as well as genes involved in RNA splicing (*SF3B1*, *SRSF2)* (Appendix A).

A total of ten *BCR-ABL*-negative MPN samples were included. Among them, four cases carried the prevalent V617F mutation in *JAK2*. One patient with a mutation at codon 515 of *MPL* was identified and one carried a deletion in exon 9 of *CALR* involving a deletion of 52 bp which belongs to the type I group of mutations. In the remaining four samples, no mutations were found either in *JAK2*, *MPL* or *CALR*. In two of the remaining four samples, that is, in two out of ten cases (20%), the Philadelphia-negative MPN diagnosis was questioned based on a negative result for *JAK2*, *MPL* and *CALR* and the centers were suggested to review the cases. In two other cases (20%), a *FIP1L1-PDGFRA* fusion was detected, which established the diagnosis of myeloid/lymphoid neoplasms with *PDGFRA* rearrangement according to the WHO 2016 classification. Therefore, the diagnosis was modified in two out of ten patients (20%) submitted as MPN based on the Haematology OncoKitDx results (Appendix A).

#### 3.2.4. Genetic Alterations in Lymphoblastic Leukemia

A total amount of 13 ALL samples were included in the study. Eleven samples belonged to B-cell ALL patients and two to T-ALL. Regarding the age of the patients, nine of them corresponded to pediatric patients and four to adults. Among the genetic alterations identified, the *ETV6-RUNX1* fusion should be highlighted, as it implies a good prognosis, and was detected in two of them (15.4%). Moreover, three patients (23.1%) carried the deletion of the *CDKN2A/CDKN2B* locus, which is associated with a bad prognosis. Two patients had the deletion of the *PAX5* gene (15.4%), with clinical implication from the diagnostic point of view, and five samples (38.5%) carried variants in RAS signaling pathway genes (*KRAS/NRAS*). *IKZF1* deletion was detected exclusively in one ALL sample (7.7%); *NOTCH1* mutations were identified only in one patient as well (7.7%). The size of the cohort has not allowed the detection of other alterations that frequently occur in ALL; however, it was possible to obtain a CNV plot for each of the 13 samples. The conventional karyotype was not available, as they were newly diagnosed samples; therefore, it was not possible to carry out a comparative study. Only one sample had a hyperdiploid karyotype (sample 12). Another one had a complex karyotype (sample 13) and 42% of the remaining samples had an aneuploid karyotype. No case of low hyperdiploidy, neither near haploid nor any intrachromosomic amplification of 21 (iAMP21), were identified. The results obtained are shown in Table 3.

The CNV plot consistent with a hyperdiploid karyotype obtained in patient number 12 is shown in Figure 3.

CNV plot obtained is consistent with hyperdiploid karyotype: 53, XXY: +5, +8, +12, +14, +17, +21, +X. Several focal chromosomal deletions are also observed in this sample.

#### 3.2.5. Potential Germline Variants

Variants of potential germline origin were detected in 33 patients (19.4%). The gene where most mutations consistent with a germline origin were detected was *TP53* (12), followed by *RUNX1* (8), *CEBPA* (6), *GATA2* (4), *ETV6* (2) and *DDX41* (1). In these cases, it was proposed to rule out their constitutional origin in the patient. Neither mutations of potential germline origin in *SRP72* nor 14q32.2 duplications were identified. In one patient diagnosed with AML (A696), a duplication in 14q32.2 was detected at chromosomal position Chr14: 98550000-98750000. However, this region does not include genes related to genetic predisposition and the medical history was not compatible with this rare entity. Therefore, it was not considered to be involved in the disease of the patient (Appendix A).

## 4. Discussion

The present work shows the analytical validation and clinical feasibility of the NGS-based panel Haematology OncoKitDx. The panel has been designed to reach an adequate molecular diagnosis, transfer prognostic information and detect alterations that may respond to targeted therapies in patients with hematological neoplasms while examining a single DNA sample. The clinical classification of the variants was adapted to current international guidelines. This panel has been optimized to detect SNVs, rearrangements and CNVs, as well as some polymorphisms related to pharmacogenetics. Due to the relevance of cytogenetic alterations in onco-hematology, the panel was specially designed to detect large CNVs throughout the entire genome. It has been analytically validated on Illumina’s NextSeq550 System platform by analyzing commercial reference DNA samples.

The protocol integrates highly sensitive capture of regions of interest with hybridization probes and molecular barcoding of each DNA fragment with a single adapter for high-performance massive sequencing (NGS). In this validation, the specific detection of the variants previously identified in the target genes has been verified, as well as the repeatability, reproducibility and detection limit of the technique. The results obtained highlight the robustness of the assay, with overall repeatability and reproducibility values greater than 99.9%, specificity more than 99% and sensibility greater than 99%. The limit of detection established for the bioinformatics pipeline was >2% for structural variants and small insertions/duplications. The limit of detection for CNV with respect to the total copies of a sample has been established at 20%. The validated bioinformatics pipeline achieves a mean coverage of 1400× and 99.3% of the target region acquires a 100× read depth.

From an analytical point of view, the robustness of the panel to detect variants with clinical implications has been confirmed. According to the information gathered in the main clinical guidelines, as well as in the previous literature [3,4,5,6,7,8,9,10,13,14], variants of diagnostic, prognostic, therapeutic and resistance to therapy value were identified in 78.8%, 81.8%, 37.1% and 30% of all patients, respectively. Furthermore, only 6% of all cases did not carry clinically relevant variants. This fact is neither a surprise, nor does it provide a specific scientific value, but it is congruent with the literature available. It simply shows that the main biomarkers included in clinical guidelines and internationally recognized articles can be adequately characterized by means of this NGS-based approach. This description is only intended to engage forward thinking on the cost-effectiveness of analyzing the high number of clinically relevant biomarkers with NGS solutions or several conventional techniques directed at specific markers. It is not intended to draw conclusions, but simply to initiate debate and raise the hypothesis that NGS technology might replace conventional techniques in onco-hematology, if its sensitivity and specificity continue to be demonstrated, the interpretation of the results is standardized and, of course, a favorable cost-effectiveness ratio is demonstrated.

The clinical feasibility of the panel has been first evaluated when classifying AML patients based on the ELN 2017 genetic risk classification. From this point of view, the results obtained in the group of 33 patients are promising. It was possible to assign a genetic risk group in cases where the karyotype could not be obtained and the results were consistent between both strategies (Commercial NGS panel plus Karyotype vs. Haematology OncoKitDx) for the remaining cases. The Cohen’s Kappa test results (K = 0.94) backs the agreement between both approximations. The only non-concordant case was due to a non-detection of two real mutations by the sequencing performed in its hospital. In a preliminary way, the DNA panel Haematology OncoKitDx might be considered a useful tool for the classification of patients by genetic risk group, at least when the karyotype cannot be obtained. However, these promising results should be confirmed in a longer series.

On another note, the Haematology OncoKitDx panel seems to be a reliable and efficient tool for establishing a WHO diagnosis in AML patients. This is supported by the fact that we have established the same WHO diagnosis as the hospitals in 97.4% of the compared AML samples. Meanwhile, to arrive at the final diagnosis, the centers used an NGS panel (with the preparation of two sequencing libraries: DNA + RNA and, at least, a conventional karyotype), while in the present study, we established the diagnosis only based on the results derived from the Haematology OncoKitDx panel. The only discrepancy was due to an event not included in the first version of the panel, which will foreseeably be detected in the second one. Concerning patients with myelodysplastic syndromes, the incidence of mutations in the main genes is consistent with that previously described in the literature. It demonstrates adequate correlation between the conventional karyotype and the CNV plot obtained by the panel to assign an IPSS-R cytogenetic risk group. The Cohen’s Kappa coefficient (K = 0.71) result was positive. However, the discrepancies found in three patients highlight some of the limitations of each of the techniques used. In patient number 17, Haematology OncoKitDx identified two CNVs not reported by the conventional karyotype. The sensitivity to detect CNVs by means of a conventional karyotype is less than the sensitivity achieved by Haematology OncoKitDx (our panel has been validated for detecting CNVs of 0.5 Kb). In order to classify the patients according to IPSS-R cytogenetic risk group, the minimum CNV size counts for the categorization should be established. On the other hand, the balanced translocations observed by conventional karyotype are not detected by means of a CNV plot obtained from next-generation sequencing. These events are detected as gene fusions by NGS techniques; therefore, this fusion must be considered as a cytogenetic event that should be accounted for when assigning an IPSS-R risk group. A limitation of DNA-based NGS would be when the rearrangement observed by conventional karyotype results in a gene fusion that is not among the targets of the panel. Although all clinically significant fusions have been included, some fusions may occur at an unusual breakpoint and not be detected by DNA-based NGS panels. This could be the case with patient number 3. This panel captures *ETV6* introns number 2, 3, 4 and 5 (NM_001987). The second version will include introns 6 and 7 as well. Finally, derived chromosomes resulting from unbalanced chromosomal events are accounted by conventional cytogenetics as a single event. However, in some cases, such as that of patient number 6, there is a gain and a loss of material from the same chromosome. Therefore, two events would be counted per NGS, while conventional karyotype would count it only as one. This may not affect the assignment of the risk group according to the IPSS-R, but it could affect other classification systems.

The analysis performed in the myeloproliferative neoplasms cohort highlights the importance of performing NGS studies for an accurate diagnosis of the different myeloid neoplasms. Since the detection of *FIP1L1*-*PDGFRA*, identified in two of the ten patients negative for *BCR*-*ABL1*, may not be detected by conventional cytogenetics, other techniques are usually essential (reverse transcriptase-polymerase chain reaction [RT-PCR], with nested RT or FISH analysis) (WHO classification). However, NGS technology may be a good strategy to test for the four most important alterations in myeloproliferative neoplasms (namely *JAK2*, *CALR*, *MPL* mutations and *BCR*-*ABL1* fusions), together with biomarkers diagnostic of myeloid/lymphoid neoplasms with rearrangements of *PDGFRA*, *PDGFRB*, *FGFR1*, and *PCM1*-*JAK2* or inherent to myelodysplastic/myeloproliferative neoplasms (MDS/MPN) diseases. By integrating all the information derived from NGS panels, a more precise diagnostic characterization is expected to be obtained in myeloid neoplasms.

Regarding acute lymphoblastic leukemia, although the series included in the validation was small, the validation has once again revealed the possibility of integrating the detection of point mutations, fusions, and CNVs of both single gene (*IKZF1*) or certain locus (*CDKN2A/B*) and large chromosomal aberrations (ploidy). This strategy has made it possible to analyze the main biomarkers with diagnostic and prognostic implications in ALL through a single analysis.

The germline implications of some of the target genes make it necessary to consider the possible constitutional origin of a large number of variants. In our study, more than 18% of all patients carried at least one variant consistent with a germline origin. The inclusion of these genes in the panel is essential, since they constitute separate entities according to the WHO 2016 classification.

## 5. Conclusions

Next-generation sequencing techniques are currently the best available way to integrate a large number of genetic and genomic biomarkers. Proper molecular categorization of patients with hematological malignancies requires testing for several genetic markers. One of the greatest challenges NGS is facing is the ability to determine CNVs of chromosomal regions. However, thanks to the quality of the sample available in the context of hematological diseases, DNA-based sequencing techniques allow for very promising approaches to conventional cytogenetics.

## Figures and Tables

**Figure 1 cancers-14-01986-f001:**
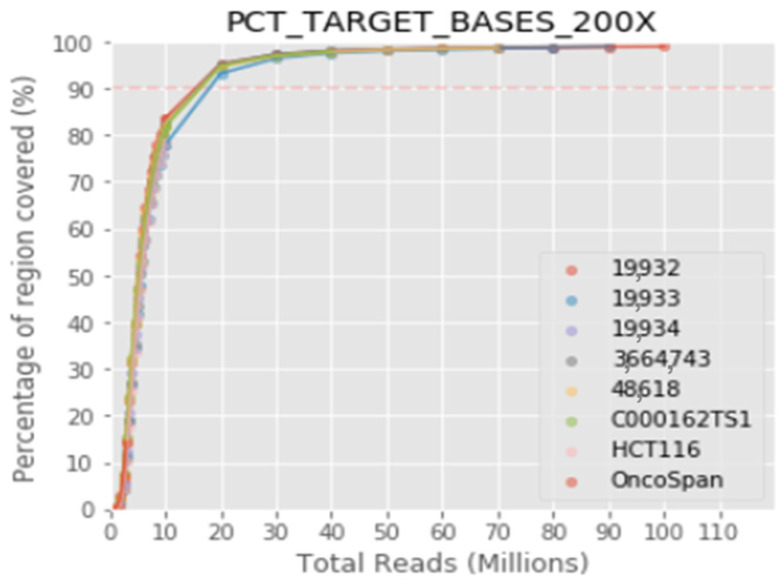
Up to 8 samples can be sequenced in the same run for the NextSeq 500/550 Mid Output v2.5 kit and 24 samples for the NextSeq 500/550 High Output v2.5 kit, ensuring a minimum of 17.5 million passing filter clusters per sample and a coverage of 97.7% with a depth of 200×, or 99.3% with a depth of 100×.

**Figure 2 cancers-14-01986-f002:**
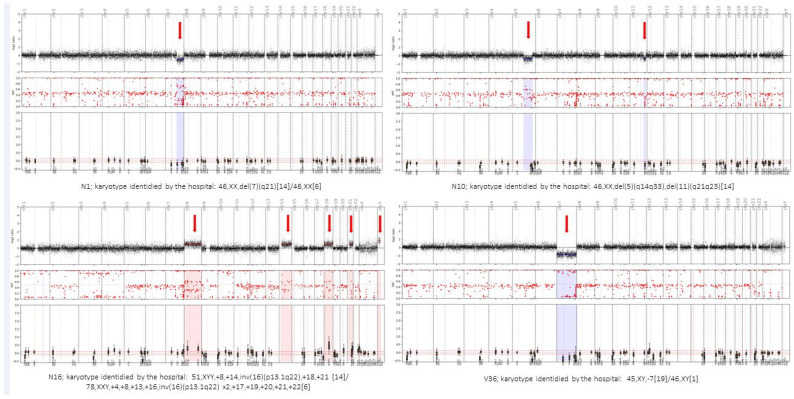
Comparative examples between the conventional karyotype reported by the hospital and the CNV plot obtained using the Haematology OncoKitDX. The arrows indicate the gains and losses of chromosomal material evidenced by the bioinformatics pipeline, which is based on the off-target of the panel. The central image of small red dots is obtained from 996 SNPs distributed throughout the genome. The loss of heterozygosity that is detected in the regions affected by a CNV reinforces the gain or loss detected by means of the off-target-based pipeline. The lower image of each example represents the target genes of the panel in their chromosomal position.

**Figure 3 cancers-14-01986-f003:**
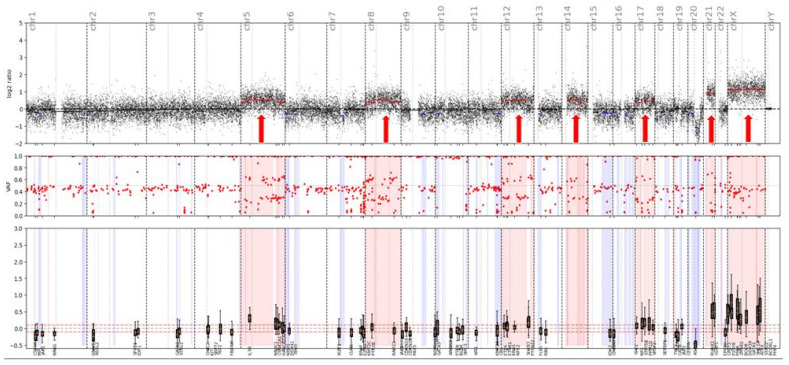
CNV plot—Patient number 12; B-ALL.

**Table 1 cancers-14-01986-t001:** Genetic alterations among 34 myelodysplastic syndromes. Number of patients carrying the variant and the corresponding percentage to this number. Prevalence of mutations observed in this series and prevalence reported in the literature.

Mutated or Rearranged Gene	Number of Patients	Percentage of Patients (%)	Overall Incidence Previous Literature (%)
*ASXL1*	11	31	15–25
*TET2*	10	28	20–25
*SF3B1*	6	17	20–30
*U2AF1*	6	17	8–12
*TP53*	5	14	8–12
*JAK2*	3	9	<5
*SRSF2*	3	9	10–15
*STAG2*	3	9	5–10
*IDH1/IDH2*	3	9	<10
*DNMT3A*	2	6	12–18
*EZH2*	2	6	5–10
*RUNX1*	2	6	10–15
*NF1*	2	6	<5
*ZRSR2*	1	3	5–10
*CBL*	2	6	<5
*SETBP1*	1	3	<5
*ETV6*	1	3	<5

**Table 2 cancers-14-01986-t002:** Cytogenetic risk assigned by conventional karyotype according to the IPSS-R classification vs. that obtained from the CNV plot generated by Haematology OncoKitDX.

Patient Number	Gender	Karyotype Identified by the Hospital	IPSS-R Cytogenetic Risk Groups	CNV Plot Identified by Haematology OncoKitDx	IPSS-R Cytogenetic Risk Groups	Hospital vs. Haematology OncoKitDx
1	Male	46,XY[20]	Good	46, XY	Good	Match
2	Male	46,XY[20]	Good	46, XY	Good	Match
3	Male	46,XY,t(3;12)(q26;p13),del(5)(q12q33),del(11)(q12)[20]	Poor	46, XY: 5qdel, 11qdel	Intermediate	Disparity
4	Male	45,XY,del(5)(q13q34),del(7)(q31),del(12)(p11),t(16;18)(q12;q11),−17[13]/46,XY,del(5)(q13q34)[7]	Very Poor	45, XY: 5qdel, 7pdel, 7qdel, 17pdel, 18del	Very Poor	Match
5	Male	46,XY[20]	Good	46, XY	Good	Match
6	Female	45,XX,−3,del(5)(q13q31),der(6)t(3;6)(q11;p25)[8]	Poor	46, XX: 3pdel, 5qdel, 6pgain, 6pdel	Very Poor	Disparity
7	Female	46,XX,del(5)(q13q31)	Good	46, XX, 5qdel	Good	Match
8	Male	46,XY[20]	Good	46, XY	Good	Match
9	Male	46,XY,del(20)(q11.21q11.31)[16]/46,XY[4]	Good	46, XY; 20qdel	Good	Match
10	Female	46,XX,del(5)(q13q31)	Good	46, XX, 5qdel	Good	Match
11	Female	45,XX,-7[12]	Poor	45, XX: 7del	Poor	Match
12	Female	45–48,XX,del(5)(q13q31),t(3;11)(p21,p15),+2mar[cp8]/45–48,XX,del(5),add(2)(q33),+1mar[cp4]/45–48,XX,del(5),+2,−3,+2mar[cp4]	Very Poor	47, XX: 1pgain, 3pdel, 5qdel, 18qdel, 19gain, 20pgain, 20qdel	Very Poor	Match
13	Male	46,XY,del(20)(q11.2q13.3)[18]/46,XY[2]	Good	46, XY: 20qdel	Good	Match
14	Female	47,XX,+8[14]/46,XX[6]	Intermediate	47, XX: 8gain	Intermediate	Match
15	Male	46,XY[20]	Good	46, XY	Good	Match
16	Female	46,XX,del(5)(q13q31)[16]/46,XX[4]	Good	46, XX, 5qdel	Good	Match
17	Male	46,XY,add(7)(p10),i(17)(q10)[20]	Intermediate	46, XY: 1pdel, 3qgain, 7pdel, i17	Very Poor	Disparity

**Table 3 cancers-14-01986-t003:** Ploidy and distribution of mutations in *TP53*, *PTEN*, *NOTCH1*, *KRAS*, *NRAS*, *FBXW7*, deletions *IKZF1*, *CDKN2A/B*, *TP53*, and gene fusions *BCR-ABL1*, *E2A-PBX1 and KMT2A* rearrangements in 13 ALL samples.

Patient Number	Diagnosis	CNV Plot Obtained	Ploidy	Prognosis Genes Mutated or Deleted
1	B-ALL	48, XXX: +22, +X	Aneuploid	*CDKN2A/B*
2	B-ALL	47, XX: 9p-, +21	Aneuploid; segmental anomaly	*CDKN2A/B*
3	T-ALL	46, XX	Diploid	*NOTCH1*
4	B-ALL	47, XY: +21	Aneuploid	*NRAS*
5	B-ALL	46, XX	Diploid	-
6	B-ALL	46, XY	Diploid	-
7	B-ALL	46, XY: 8q+, 9p- 12p-	Diploid; segmental anomalies	*NRAS*, *CDKN2A/B*
8	T-ALL	46, XY: 4q+, 7p-	Diploid; segmental anomalies	-
9	B-ALL	46, XY	Diploid	-
10	B-ALL	47, XY: 13q-, 20q-, +21	Aneuploid; segmental anomalies	*KRAS*
11	B-ALL	48, XX: +10, +21	Aneuploid	*NRAS*
12	B-ALL	53, XXY: +5, +8, +12, +14, +17, +21, +X	Hyperdiploid (>50 chromosomes)	*KRAS*
13	B-ALL	46, XY: 7p-, 8p-, 8q+, 9p-, 11p-, +22	Complex karyotype	*IKZF1*, *CDKN2A/B*

## Data Availability

All clinical and genetic information derived from the study was included in the paper itself, as well as in the Appendix A.

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
