# Peer review of "Next-Generation DNA Sequencing-Based Gene Panel for Diagnosis and Genetic Risk Stratification in Onco-Hematology"

_cancers, 2022, doi:10.3390/cancers14081986_

Round 1

Reviewer 1 Report

Cancers-1564426-peer-review-v1Cancers

Authors have comprehensively profiled onco-hematological diseases including AML, MPN, MDS and ALL through a well designed, analytically validated molecular diagnostic kit in context to clinical relevance named as “Haematology OncoKitDx panel”. The kit includes coding regions of 78 genes for single nucleotide variants (SNVs), small insertions or deletions (INDELs); fusions in 27 target genes; specific probes and custom bioinformatics for CNVs, inclusion of germline variants, 14 relevant SNPs in pharmacogenetics GKB database. Authors have classified their design into analytical and clinically relevant validations as well as genetic-risk assessment.

I-Analytical Assay validation: which included a total of 63 samples in which 54 were Bone Marrow samples from patients and 6 were commercial reference samples

19 samples-for sensitivity and specificity for SNVs and small indels

12 samples-for CNV evaluation

19 samples-for fusion and large rearrangements

3 samples-for pharmacogenetics testing

11 samples -for FLT3-ITD detection analysis

II-Both repeatability and reproducibility were conducted with samples previously analyzed in the above technical validation

III- clinical feasibility validation

170 bone marrow samples for SNVs, INELs, CNVs and fusions

Findings:

  • The analytical and clinical feasibility validation revealed a sensitivity and specificity close to 100% for all the parameters evaluated, considering a detection limit of 2% in SNVs and SVs, as well as 20% for CNVs.
  • Clinically relevant mutations were detected in 94% of all patients.
  • IV The correlation between the genetic risk classification of AML (according to ELN 2017) established by the hospitals and that obtained by the Hematology OncoKitDx panel showed an almost perfect correlation (K = 0.94).
  • Categorized samples in different categories<0-Poor; <0-<02-Slight;<0.2-<0.4-Fair;<0.4-<0.6-0.8-Substantial;0.8-1-almost perfect;1-Perfect

Hence, the above “Haematology OncoKitDx panel” with good sequencing quality scores Q30 = 92.9%; average percentage of clusters passing filters were 90%; Mean coverage was 1400X (98.41% of the bases had>20% of the average coverage and 97.7% of bases has achieved a depth of 200x) suggests a robust panel for profiling onco-hematological diseases. This statement was very well proved by following examples

  • For 33 of 98 AML cases, the genetic results obtained by means of other commercial NGS panels, as well as from the conventional karyotype, were available and could be provided by the hospitals. The ELN 2017 risk group assigned by the different hospitals was contrasted with the risk group obtained by the Hematology OncoKitDx panel. 

  • 5 of the 33 patients (15%), the karyotype result provided by their hospitals was inconclusive and a risk group could not be assigned. The Hematology OncoKitDx panel was able to assign a risk group in these five cases: N-4, M-7, M-11, M-12, V-16.
  • In 27 of the remaining 28 patients (96.4%), the risk group assigned by the reference hospital was like the one assigned by Hematology OncoKitDx
  • 1 of 28 cases (N-22; 3.6%), Hospital recommended it in the Intermediate group while the Hematology OncoKitDx panel assigned it an adverse risk according to the ELN 2017 guideline.

I appreciate the authors well-designed, analytically, and clinically validated robust panel for onco-hematological databases. 

Author Response

Thank you very much

Reviewer 2 Report

Authors constructed gene panels based on NGS by using the abnormalities of 103 genes. The results of this study are excellent. There no points to be revised.

Author Response

thank you very much

Reviewer 3 Report

In this manuscript, Gargallo and Co-authors report the development of a new NGS panel for genetic risk stratification in onco-hematology. This is a relevant, and needed, approach to onco-hematology. Authors present this developed panel as a new alternative to the laborious and time-consuming conventional technologies. Although the analytical validation revealed a high specificity and sensitivity of the NGS-panel, some questions are raised as follows:

  • The challenges of NGS implementation in clinical scenario are far more complex than the different nature of the biomarkers to be analyzed. A possible disadvantage of the developed NGS panel is the interpretation of the data. This should be discussed in the text.
  • In the text it is not clear whether blind controls from hematological patients are correctly diagnosis by this new panel. Indeed, it is not understandable whether it is possible to distinguish for instances an AML from a MDS patient, as well as the expertise of the clinician to do so with this panel. These are important parameters to be considered in measuring clinical feasibility.
  • In line with the previous comment, it is also not clear the advantages of using a kit to detect multiple hematological malignancies. Probably the most rational approach will be to have a panel for each disease.
  • Regarding supplementary material 1, it is important to understand, at least for AML samples, the status of the disease, if they were collected at diagnosis, relapse, MRD, etc.
  • List the genes grouped in Tier I and II.
  • ALL and MPN cohorts are too small and the diseases too complex to be possible to report conclusions.
  • According to supplementary material 1, almost all variants were considered of prognosis or diagnosis for AML and MDS and of therapy and diagnosis for MPN. Thus, it is not strange that these variants represent the higher percentages in the respective diseases. This is a completed biased analysis that does not make much sense. The text needs to clarify what is the point with section 3.1.1.
  • Table 1 reports the percentage of frequency of different mutations in the AML cohort of 98 patients. The objective of presenting this table is also not clear. If it is to show that the developed panel allows diagnosis according to WHO classification system, it has to be presented in a different way.
  • A relevant aspect that is not clear in the text is the type of mutations appearing in the same gene.
  • This seems to be a good NGS panel for risk stratification but this is not the most clinically relevant classification. 

Author Response

Thank you very much. 

In this manuscript, Gargallo and Co-authors report the development of a new NGS panel for genetic risk stratification in onco-hematology. This is a relevant, and needed, approach to onco-hematology. Authors present this developed panel as a new alternative to the laborious and time-consuming conventional technologies. Although the analytical validation revealed a high specificity and sensitivity of the NGS-panel, some questions are raised as follows:

  • The challenges of NGS implementation in clinical scenario are far more complex than the different nature of the biomarkers to be analyzed. A possible disadvantage of the developed NGS panel is the interpretation of the data. This should be discussed in the text: thank you very much; We agree with you at this point. We have introduced some sentences within the introduction in order to remark some challenges:

Certainly, implementing NGS-based gene panels in clinical practice raises several challenges. Some of them are inherent to NGS technologies and could be solved with experience and knowledge. However, two particular challenges hamper the implementation of NGS solutions in daily clinical practice of onco-hematology laboratories. First, the current international guidelines are based on previous experience; therefore, NGS, due to a lack of standardization, is not the main technology recommended for analyzing several biomarkers. Second, there are insufficient cost-efficiency studies in this clinical field that would demonstrate the advantage of implementing NGS.

Despite these challenges, NGS-based gene panels enable the identification of target genetic alterations from a single sample and in a single assay. Therefore, it may be a useful strategy for molecular diagnosis in onco-hematology that could replace or complement the conventional techniques.

  • In the text it is not clear whether blind controls from hematological patients are correctly diagnosis by this new panel. thank you for your comments and we absolutely agree with you at this point. It is true that we have important information about WHO 2016 diagnosis established by centers and by Haematology OncoKitDx and it was not included in the previous version. We have added some sentences presenting the collected data:

Finally, the molecular diagnosis based on the WHO 2016 classification established by the hospital was communicated in 39 cases (39/98). The same diagnosis was established by means of sequencing Haematology OncoKitDx in 38 of these 39 cases. The only difference was evidenced when analyzing the F-1 sample, whose diagnosis provided by the center was AML with inv(3)(q21.3q26.2) or t(3;3)(q21.3;q26.2); GATA2,MECOM(EVI1). However, the first version of the Haematology OncoKitDx panel presented in this work was not capable of detecting said molecular event, thus, being an expected disagreement (supplementary material 2).

  • Indeed, it is not understandable whether it is possible to distinguish for instances an AML from a MDS patient, as well as the expertise of the clinician to do so with this panel. These are important parameters to be considered in measuring clinical feasibility.: Thank you very much for your comment. This is an important point that requires clarification. The present work shares the analytical validation and certain aspects of its clinical feasibility. However, this study has been carried out by the company collaborating with researchers from different hospitals, but none of them is a customer. Therefore, none of them has implemented the Haematology OncoKitDx kit in their hospitals. Accordingly, a complete clinical feasibility evaluation in their hands could not be performed. Such an evaluation will be hopefully possible in the near future.
  • In line with the previous comment, it is also not clear the advantages of using a kit to detect multiple hematological malignancies. Probably the most rational approach will be to have a panel for each disease.

This point includes ad very important questions that need to be answered. For the purpose of designing and developing this gene panel, multiple working parameters were taken into account in order to decide on the best solution. Our company is an Agilent technologies partner, and therefore, we employ the Magnis NGS Prep System. Automation by means of using Magnis does not allow combining the preparation of libraries from different gene panels. Likewise, it is designed to work with 8 samples at the same time. Thus, in the event of working with four different panels, hospitals would have to collect 8 AML, 8 MDS, 8 ALL... in order to complete an automated sequencing run for every panel. Keeping in mind that the number of patients is not so high in most hospitals, and that the molecular results in acute leukemia are urgently needed, the best way to group 8 samples and to sequence them quickly and at the best price, is by using a panel that integrates different diseases. Moreover, we must not forget that although each of the biomarkers is of special interest for one of the pathologies, many of them add clinical information to be transferred to other diseases, although the evidence may be relatively lower. In other words, a significant number of genes are of interest in various pathologies, and therefore, it seems appropriate to try to group genes and pathologies within the same panel design.

  • Regarding supplementary material 1, it is important to understand, at least for AML samples, the status of the disease, if they were collected at diagnosis, relapse, MRD, etc.

It is true, we did not include this detail when introducing the patients. We have added these two points:

Regarding newly diagnosed AML patients line 304

Concerning MDS patient samples studied at diagnosis line 307

  • List the genes grouped in Tier I and II: Thank you. We agree with you, it might be clarifier. However, every gene has a different tier depending on the disease and the clinical aspect analyzed (Diagnosis, prognosis, resistance or therapy). We thought that this type of table is a bit complex and we decided not to implement it. We decided to incorporate this information, implicitly, in supplementary material 1. If you don’t mind, we think that it could be enough for this paper.

  • ALL and MPN cohorts are too small and the diseases too complex to be possible to report conclusions: thank you for this comment. We think the same about these two cohorts: we valued the possibility of not introducing comments related to both pathologies, but finally, we decided to include a few descriptive comments, but without drawing important conclusions.
  • According to supplementary material 1, almost all variants were considered of prognosis or diagnosis for AML and MDS and of therapy and diagnosis for MPN. Thus, it is not strange that these variants represent the higher percentages in the respective diseases. This is a completed biased analysis that does not make much sense. The text needs to clarify what is the point with section 3.1.1.:

After assessing your comment, we also perceive that section 3.1.1 may generate confusion and a biased view of reality. To avoid an erroneous impression of this section meaning, we have incorporated a clarification paragraph in the material and methods section, as well as other one in the discussion.

Line 183:

The percentage of clinically relevant variants grouped into each of the four parameters of interest (diagnosis, prognosis, therapy and resistance) for each neoplasm aims to describe and summarize the collected data in a global way. As the panel includes the most frequently mutated genes and genes with the highest level of clinical evidence for each of the pathologies analyzed, the number of variants with clinical evidence at the diagnostic, prognostic, therapeutic and resistance-to-therapy levels is expected to be high and consistent with previous reports. This description is not intended to create the impression that the panel detects something other than what might be expected based on the knowledge available on the analyzed pathology; it is simply to illustrate how the biomarkers that are expected to be found may be detected by a single test, in this case by this NGS test.

Line 510:

This fact is neither a surprise, nor it provides a specific scientific value; it simply shows that the main biomarkers included in clinical guidelines and in internationally recognized articles can be adequately characterized by means of this NGS-based approach. This description is only intended to engage forward thinking on the cost-effectiveness of analyzing the high number of clinically relevant biomarkers by using NGS solutions or several conventional techniques directed at specific markers. It is not intended to draw conclusions, but simply to initiate debate and raise the hypothesis that NGS technology might replace conventional techniques in onco-hematology, if its sensitivity and specificity continue to be demonstrated, the interpretation of the results is standardized and, of course, a favorable cost-effectiveness ratio is demonstrated.

  • Table 1 reports the percentage of frequency of different mutations in the AML cohort of 98 patients. The objective of presenting this table is also not clear. If it is to show that the developed panel allows diagnosis according to WHO classification system, it has to be presented in a different way: We agree with you. This table is a bit poor. We have removed it and modified the numeration of all other tables.

  • A relevant aspect that is not clear in the text is the type of mutations appearing in the same gene.: It is true, we did not speak about this point in the manuscript. We have classified the variants according to ACMG criteria strictly throughout the process. Therefore, the variants detected and considered pathogenic or likely pathogenic have been previously described or the type of mutation explains the classification (for example a loss of function variant in a gene commonly considered a gene supresor in a concrete pathology). We considered that the inclusion of this point would not be scientifically relevant. Our opinion was that this point would extend the text, but it would not contribute too much to the paper.

  • This seems to be a good NGS panel for risk stratification but this is not the most clinically relevant classification: Ok, thank you; we understand your opinion and in fact we share it. The clinical performance of this panel should be evaluated with more samples, in other hands and exploring extensively different clinical aspects. However, we also think that the data related to genetic risk classification based on ELN 2017 and WHO diagnosis in AML established from this kit is very interesting. Moreover, the great correlation between cytogenetics and CNV plot from NGS sequencing is valuable and it warrants to be shared by this way.

We have reviewed again the English and we hope it will be enough.

Round 2

Reviewer 3 Report

Authors have made some alterations in the manuscript, nevertheless, proper attention was not given to some raised questions, as detailed below.

- In the first revision, it was not clear whether blind controls from hematological patients are correctly diagnosis by this new panel. Indeed, it was not understandable whether it is possible to distinguish for instances an AML from a MDS patient, as well as the expertise of the clinician to do so with this panel. The authors did not address this concern and it is quite relevant to understand the feasibility and utility of the panel.

- The title of the manuscript indicates a NGS based panel for diagnosis and genetic risk stratification in onco-hematology. Onco-hematology is far more than AML and MDS, which authors do not know it the panel differentiates or the advantages of the panel. Furthermore, in the keywords several other diseases within onco-hematology are mentioned and it is important to be objective and clear about the panel that was developed and validated.

- The issues raised regarding the variants of prognosis or diagnosis for AML and MDS and of therapy and diagnosis for MPN, were not properly addressed. Thus, it is not strange that these variants represent the higher percentages in the respective diseases. This is a completed biased analysis that does not make much sense. The text needs to clarify what is the point with section 3.1.1. the paragraph that was introduced does not solve the issues.

Author Response

thank you 

Round 3

Reviewer 3 Report

The manuscript by Gargallo P. and the team has been substantially improved. There are still some minor aspects that need Authors' attention as follows:

  • Figure 2 still needs to be improved or removed. As mentioned, before it does not add nothing to the conclusions.
  • Explain better the sentence on line 243.

Author Response

thank you
